# Age-Stage, Two-Sex Life Table and Functional Response of *Amblyseius orientalis* (Acari: Phytoseiidae) Feeding on Different Nutrient Sources

**DOI:** 10.3390/insects13110983

**Published:** 2022-10-26

**Authors:** Keyu Pan, Tianrong Xin, Yibing Chen, Hongyan Wang, Kexin Wen, Yimeng Liu, Zhenzhen Li, Zhiwen Zou, Bin Xia

**Affiliations:** School of Life Sciences, Nanchang University, Nanchang 330031, China

**Keywords:** *Amblyseius orientalis*, predatory mite, age-stage two-sex life table, nutrient sources, biological control

## Abstract

**Simple Summary:**

*Amblyseius orientalis* is a predatory mite that belongs to the family Phytoseiidae. It is mainly found in Jiangxi, Shanghai, Guangdong, and other areas of China. *A. orientalis* is a dominant predatory mite species in China and is also important for agricultural development and biological control. Thus, research on *A. orientalis* is necessary. We conducted experiments to determine the growth, development, reproduction, and functional response of *A. orientalis* in this study by indoor single-head rearing at 25 ± 1 °C, 65 ± 5% relative humidity, and a photoperiod of 16 h:8 h light/dark cycle under laboratory conditions. Through experiments, we finally determined that pollen + yeast + sucrose treatment was the most favorable for captive breeding of *A. orientalis*. The results of the study provided a theoretical basis for indoor rearing, propagation, and utilization of *A. orientalis*.

**Abstract:**

*Amblyseius orientalis* Ehara is a predatory mite that belongs to the family Phytoseiidae. It is mainly found in Jiangxi, Shanghai, Guangdong, and other areas of China. Although *A. orientalis* is a dominant predatory mite species in China and is also important for agriculture and biological control, not many studies have investigated it. Thus, research on *A. orientalis* is necessary. However, its application in biological control is hindered by the absence of techniques for the mass rearing of *A. orientalis* in captivity. We conducted experiments to determine the growth, development, reproduction, and functional response of *A. orientalis* in this study by indoor single-head rearing at 25 ± 1 °C, 65 ± 5% relative humidity, and a photoperiod of a 16 h:8 h light/dark cycle under laboratory conditions. The results of the age stage, two-sex life table showed that the individuals in the pollen + yeast and pollen + yeast + sucrose groups had significantly higher oviposition period, fecundity, net reproductive rate (R_0_), and gross reproduction rate (GRR) than those in the pollen group. The results of the function response showed that the pollen + yeast + sucrose group was the most favorable for captive breeding of *A. orientalis* and had the best predatory ability along with rejuvenation and recovery ability. The results of the study provided a theoretical basis for indoor rearing, propagation, and utilization of *A. orientalis*.

## 1. Introduction

Predatory mites are widely distributed in different parts of the world and are used for biological control. The application of predatory mites has a significant impact on agriculture, horticulture, forestry, etc., and the use of predatory mites for biological control is better for the environment and human health compared to the use of chemical methods. They are effective predatory biological control agents with short developmental periods, high reproductive capacity, wide geographical distribution, and strong predatory ability [1,2]. The difficulty in large-scale rearing is the main hurdle associated with the application of predatory mites, and only a few species of mites, such as *Neoseiulus barkeri* and *N. cucumeris*, can be propagated efficiently and inexpensively.

*Amblyseius orientalis* is a predatory mite belonging to the Phytoseiidae family and is found primarily in Jiangxi, Shanghai, Guangdong, and other parts of China [3]. It is a dominant predatory mite and primarily preys on spider mites, such as *Amphitetranychus viennensis* and *Panonychus citri* [4]. However, low-cost mass propagation of *A. orientalis* is still not possible, and artificial mass-rearing techniques for *A. orientalis* still need significant improvements, thus limiting its application in biological control to some extent.

Mite reproduction is highly dependent on nutrients. In nature, mites receive nutrients from sap secreted by plant blossoms and leaves, pollen, and honeydew excreted by microscopic insects such as aphids [5]. These nutrients help in the growth and maturation of reproductive organs, enhance egg production, and help populations thrive and reproduce [6]. Proteins and sugars affect mite reproduction. Supplementing the diet of phytoseiid mites with substances containing high levels of protein and sugar significantly affects their growth and development. Huang He et al. found that feeding yeast, glucose, and white granulated sugar to *N. barkeri* shortened their development by up to three days. Female mites in the treatment group produced significantly more eggs than those in the control group. In the white granulated sugar group, the developmental period of the mites was shortened to five days. The addition of yeast, glucose and sugar can significantly enhance the net rate of increase in reproductive parameters of the population compared to the changes in reproductive parameters of the control group; the addition of yeast resulted in the highest increase in the endogenous growth rate of the *N. barkeri* population.

In addition to elucidating the effects of nutrients on the growth, development and reproduction of *A. orientalis*, the multifaceted effects of nutrients on predatory mites were also investigated in a study in which changes related to predation on *Tetranychus cinnabarinus* and *Frankliniella occidentalis* by females of *N. barkeri* feeding on *Tyrophagus putrescentiae* (wheat bran and wheat bran + yeast feeding) were determined [7]. *Neoseiulus barkeri* had a Holling type II predatory functional response to both prey, and the predation of yeast-added *N. barkeri* on the prey increased significantly. Thus, the addition of yeast powder to the basal diet can increase pest predation [8].

Nutrient selection is an important factor in the rearing and application of predatory mites [9]. Some studies have expanded the population of *A.swirskii* by adding artificial diets or factitious prey and putting it into production and application [10]. Research shows *A.orientalis* is also a generalist and will prey on food such as pollen [3]. However, there is a lack of research on the growth, reproduction and predation of *A.orientalis* using nutrient sources. Therefore, the distribution and application of *A. orientalis*, as well as the development of agriculture and biological control in China, are necessary [11]. Our objective was to evaluate the effect of different food types on the growth, development, reproduction and predation ability of *A. orientalis* to find the optimal food for the laboratory culture of this species.

## 2. Materials and Methods

### 2.1. Mite Cultures

Citrus leaves were planted in pots and kept at 25 ± 5 °C, 65 ± 10% RH, and under natural light in a conservatory to raise the spider mite *Panonychus citri*, the natural prey of *A. orientalis*. Spider mites were first collected from a citrus orchard in the Nanchang University Botanical Garden that was never treated with pesticides or acaricides.

In 2021, the predatory mite *A. orientalis* was collected from a citrus orchard in Yangmei Village, Nanfeng County, Fuzhou City, Jiangxi Province, and cultured in an artificial climate box with a 16 h:8 h light/dark cycle at 25 ± 1 °C and 65 ± 10% RH.

### 2.2. Treatment Groups

In total, three treatments were designed to carry out this experiment. The pollen group was reared on rape pollen harvested from the Biological Garden of Nanchang University. The yeast extract (FMB grade) and sucrose (molecular biology grade) were purchased from the Sangon Biotech Company. In the pollen + yeast group, pollen and yeast were mixed in a 2:1 ratio. In the pollen + yeast + sucrose group, pollen, yeast, and sucrose were mixed in a 2:1:1 ratio (each individual was fed about 1 g per day).

### 2.3. Life Table Study

Individuals of *A. orientalis* were divided into three groups based on their diet, i.e., the pollen group, the pollen + yeast group, and the pollen + yeast + sucrose group. After being reared for three generations, 50 adult female mites of similar size were selected from each group for the experiment. The studies were carried out at 25 ± 1 °C and 65 ± 10% RH, maintaining a photoperiod of a 16 h:8 h light/dark cycle in an acclimation chamber [12,13].

Subsequently, 50 eggs were collected from the third-generation females. Eggs were individually transferred to the experimental units, which were composed of two transparent acrylic sheets (50 mm × 50 mm × 4 mm and 50 mm × 50 mm × 1 mm) with a hole in the middle (diameter 2 cm). After the larvae emerged, the predators were given appropriate nutrient sources. The experimental units were checked twice a day to record the time taken for the life stages to develop and whether the individuals survived. Following the emergence of adults, each female was paired with two males and transferred to a separate experimental unit, where they were fed the same food as before. Daily observations were made until the end of the experiments to acquire data on adult longevity, fecundity, and survival [14]. Experiments were repeated three times for each treatment group at the same time.

### 2.4. Functional Response Study

Functional responses were also grouped in the same way as in the previous life table, and then 20 adult female *A. orientalis* were collected from each group for the experiment. Each *A. orientalis* female received different densities of *P. citri* in groups. Two transparent acrylic plates (50 mm × 50 mm × 40 mm and 50 mm × 50 mm × 1 mm) with a hole (2 cm in diameter) in the center served as experimental units for functional response tests. Additionally, five densities of adult *P. citri* mites (i.e., 4, 8, 12, 16, and 20 individuals) and five replicates for each density were used to record their functional response. Each set-up received one adult female *A. orientalis*, and the number of prey devoured was determined after 24 h [14].

A rejuvenation experiment was also conducted in which the experimental group, which fed on seven generations, was reared with *P. citri* and continued to feed for one, three, and five generations, and daily predation of *P. citri* by *A. orientalis* was recorded. The conditions used for these experiments were identical to those used for the life table trials (i.e., at 25 ± 1 °C, 65 ± 10% RH, and with a photoperiod of a 16 h:8 h light/dark cycle).

### 2.5. Data Analysis

#### 2.5.1. Age-Stage, Two-Sex Life Table

Parameters were analyzed on the basis of the age-stage, two-sex life table theory [13,15] using a TWOSEX-MSChart-2021-12-01 [16] (http://140.120.197.173/Ecology/prod02.htm, accessed on 15 October 2022). The net reproductive rate (R_0_), the mean generation time (T), the intrinsic rate of increase (r_m_), the finite rate of increase (λ) and the gross reproduction rate (GRR) were calculated as follows [17]:R_0_ = Σl_x_m_x_
T = lnR_0_/r
r_m_ = Σe^−r(x+1)^l_x_m_x_
λ = exp(r)

The age-specific survival rate (*l_x_*) (probability that a newly laid egg will survive to age *x*), the age-specific fecundity curve (*m_x_*) (the average fecundity of the individuals), the age-stage-specific survival rate (*S_xj_*) (probability that a newly laid egg will survive to age *x* and stage *j*), the age-stage life expectancy (*e_xj_*) (expected time that an individual of age *x* and stage *j* is expected to live), and the age-stage-specific fecundity (*f_xj_*) (number of hatched eggs produced by a female adult at age *x*, where *j* is the life stage number (*j* = 5)) were calculated as follows [18]:$$l_{x}=\overset{\beta }{\underset{j=1}{\sum }}S_{xj}$$
$$m_{x}=\frac{\sum_{j=1}^{\beta }S_{xj}f_{xj}}{\sum_{j=1}^{\beta }S_{xj}}$$
$$e_{xj}=\overset{\infty }{\underset{i=x}{\sum }}\overset{\beta }{\underset{y=j}{\sum }}S_{iy}^{\prime }$$

Standard errors were estimated by bootstrapping with 100,000 trials [15]. Figures were plotted using Sigmaplot v12.5 software (Systat Software; San Jose, CA, USA), and the statistical significance of the observed differences was determined using TWOSEX-MSChart software. The t score (Student’s *t*-test), degrees of freedom, eggs laid per female per day, pupal weight and survival rate (analyzed after arcsine square root transformation, formula: ARSIN (SQRT (A1 × 180/3.1415926)), at each stage until emergence, were analyzed using SPSS version 22.0. (SPSS Inc.; Chicago, IL, USA).

#### 2.5.2. Functional Response and Searching Rate

With an attack rate of (u, v) and a handling time of h, predation follows a Holling type II functional response. Holling’s type II disc equation [19] is mentioned below:$$N_{a}=\frac{aTN_{0}}{1+aT_{h}N_{0}}$$

The search rate *S* is computed using the following formula:$$S=\frac{a}{1+aT_{h}N_{0}}$$

Here, *N_a_* indicates the total number of prey killed by predators in a specific period (*T*: 24 h). For a prey item, *N*_0_ indicates the density of the prey, a indicates the attack rate, and Th indicates the predator handling time. Figures were plotted using Sigmaplot v12.5 software (Systat Software; San Jose, CA, USA).

## 3. Results

### 3.1. Age-Stage, Two-Sex Life Table

The developmental duration of the individuals in the pollen + yeast + sucrose group and the pollen + yeast group was shorter than that of the individuals in the pollen group, regardless of the females, males or the entire population (Table 1). Female and male adults survived significantly longer in the pollen + yeast + sucrose group than in the other two groups. Females in the pollen + yeast + sucrose had the longest total life span, while males in the pollen + yeast + sucrose and pollen + yeast groups had a significantly longer total life span than the males in the pollen group. Adult survival and total life span were not significantly different between the pollen + yeast + sucrose and pollen + yeast groups but were significantly more in these two groups than in the pollen group.

Individuals in the pollen + yeast and pollen + yeast + sucrose groups had significantly higher net reproductive rates (R_0_) and gross reproduction rates (GRR) than those of the pollen group (Table 2) [20,21]. The intrinsic rate of increase (r) and the finite rate of increase (λ) did not vary significantly between the three groups. The r and λ values of the *A. orientalis* populations were >0 and >1 for the three groups (Table 2), which indicated that the *A. orientalis* individuals could survive under different conditions. The individuals in the pollen group had a significantly shorter mean generation time than those in the other two groups [22].

The oviposition period of the individuals in the pollen + yeast group and the pollen + yeast + sucrose group was significantly longer than that of the individuals in the pollen group (Table 3). The fecundity of the individuals in the pollen + yeast group (25.86) and the pollen + yeast + sucrose group (27.00) was also significantly higher than that of the individuals in the pollen group (13.50). The total preoviposition period (TPOP) and the female ratio of the offspring in the three groups did not differ significantly. The adult pre-oviposition period (APOP) was significantly shorter in the pollen group than in the other two groups. No significant differences in the female ratio of the offspring were found among the three groups [23].

Age differentiation and overlap were found in all three groups (Figure 1). The life span of the males was significantly shorter than that of the females. The S_xj_ of *A. orientalis* males and females from egg to adult in the pollen group was 0.29 and 0.53, respectively; the pollen + yeast group was 0.27 and 0.47, respectively, and the pollen + yeast + sucrose group was 0.27 and 0.47, respectively, indicating that there were statistically significant differences between the sexes based on the food provided.

The l_x_ of the three groups decreased with age, and the f_x_, m_x_, and l_x_*m_x_ values of the pollen group reached a maximum on the 13th day (1.35, 0.90, and 0.71) and on the 18th day for the pollen + yeast group (1.57, 1.00, and 0.73) and the pollen + yeast + sucrose group (1.71, 1.09, and 0.80) (Figure 2). These maximums were lower than those of the pollen + yeast + sucrose group. Thus, a diet of pollen + yeast + sucrose was more conducive to the development and reproduction of *A. orientalis*. Furthermore, fluctuations in the fecundity curve suggested that the emergence and oviposition of *A. orientalis* individuals did not occur at a specific age.

In all three treatment groups, the value of e_xj_ decreased (Figure 3). For the pollen group, the pollen + yeast group, and the pollen + yeast + sucrose group, the maximum mean lifetime values were 19.05, 28.13, and 31.88 days, respectively.

At age zero (v_0.1_), the v_xj_ of the groups of *A. orientalis* individuals that were fed pollen, pollen + yeast and pollen + yeast + sucrose was 1.14, 1.15, and 1.15, respectively; the values were close to λ (Figure 4). The maximum value of the v_xj_ curve increased with age and developmental stage, reaching its highest point on the 10th day for individuals in the pollen group (6.68), the 13th day for those in the pollen + yeast group (8.69), and the 14th day for individuals in the pollen + yeast + sucrose group (8.74). Female adults reared on pollen + yeast + sucrose had the highest v_xj_.

### 3.2. Functional Response

The functional response of *A. orientalis* to *P. citri* in various treatments was of the Holling type II variety. The functional response data that fits the Holling disc equation is shown in Table 4.

The results indicated that in the same generation, the intensity of predation of *P. citri* by *A. orientalis* differed between the three treatment groups and was in the order: pollen + yeast + sucrose group > pollen + yeast group > pollen group. The instant attack rate, the predation capacity, and the daily maximum predation number of *A. orientalis* decreased from G3 to G5, and the handling time increased. From the rejuvenation group onward, the instant attack rate, predation capacity, and daily maximum predation number increased with each generation, and the handling time decreased.

All three groups initially showed a decrease in predation ability as generations progressed and then showed an increase in predation ability with generation after restrengthening (Figure 5). In each generation, individuals in the pollen + yeast + sucrose group had a higher predatory capacity than those in the other two groups, and their recovery of the predatory capacity after rejuvenation was also the best.

The search effect decreased with an increase in density for the same treatment conditions and generations and followed the pattern of pollen group < pollen + yeast group < pollen + yeast + sucrose group (for the same treatment conditions and generations) (Table 5).

The search rate was negatively correlated with the density of the prey (Figure 6). As the density of the prey increased, the value of the search effect decreased, i.e., the difficulty of finding the prey in a given space decreased with an increase in the density of the prey.

In the pollen group, the search effect decreased across generations from G3 to RG1 and increased from RG3. The pollen + yeast group had higher values than the pollen group, and the search effect started to increase in RG1, with an earlier trend than the pollen group. The pollen + yeast + sucrose group had the highest value for the search effect, which started to increase significantly in RG1 and reached its maximum value in G5 (0.92).

## 4. Discussion

Two tested types of food, the pollen + yeast + sucrose treatment and the pollen + yeast treatment, were more favorable for the development and reproduction of *A. orientalis* than the pollen treatment. The predatory ability of the individuals in the pollen + yeast + sucrose group was better than that of the individuals in the other two groups.

The R_0_, r, and λ of the pollen + yeast + sucrose group and the pollen + yeast group were not significantly different. R_0_ and GRR were significantly lower in the pollen group than in the other two groups, which showed that the food rich in protein and carbohydrates caused improvements (Table 2). The pollen + yeast + sucrose treatment and the pollen + yeast treatment were found to be more conducive to the growth and development of *A. orientalis* than the pollen treatment based on these parameters. This result is similar to previous studies on food rich in protein and carbohydrates and the growth, development and reproduction of mites [20,21]. This was probably because supplementing the diet with nutrients can improve the quality of Phytoseiidae mites. The addition of yeast, glucose, and white sugar to the diet of *Tyrophagus putrescentiae* individuals strongly affected their growth [22]. A variety of nutrient additives may be more conducive to the growth, development and reproduction of predatory mites than a single kind of nutrient additive. Therefore, we can consider adding a variety of nutritional additives to promote the growth, development and reproduction of predatory mites when we feed them artificially [23,24].

All three groups showed a decrease in predation ability across generations (Figure 5). This could be due to a decrease in the ability of predatory mites to feed over time due to the use of artificial feed. However, after rejuvenation, individuals in the pollen + yeast + sucrose group showed the best recovery from predatory ability (Figure 5). This result is similar to previous studies. The addition of yeast to the basal diet of *Neoseiulus barkeri* showed an increase in the instant attack rate and the daily maximum events of predation in the larvae of *Tetranychus cinnabarinus* and *Frankliniella occidentalis*, and a decrease in handling time [25]. The nutritional supplements in the pollen + yeast + sucrose group had less effect on the decline of the predation ability of *A. orientalis*, and the recovery ability after feeding with *P. citri* was also better, indicating that the pollen + yeast + sucrose group reduced the effect of artificial feeding on the decline of *A. orientalis* predation efficiency [18]. Furthermore, the ability to recover predatory function after eating natural prey was also better, which is more conducive to the application of *A. orientalis*.

In conclusion, *A. orientalis* individuals that were fed pollen + yeast + sucrose had the best population growth parameters, predatory ability, and rejuvenation capacity. Nutrient selection is an important factor in the rearing and application of predatory mites by humans [26]. This study established a theoretical foundation for the rearing and propagation of *A. orientalis* under controlled conditions, as well as its application in pest control.

## Figures and Tables

**Figure 1 insects-13-00983-f001:**
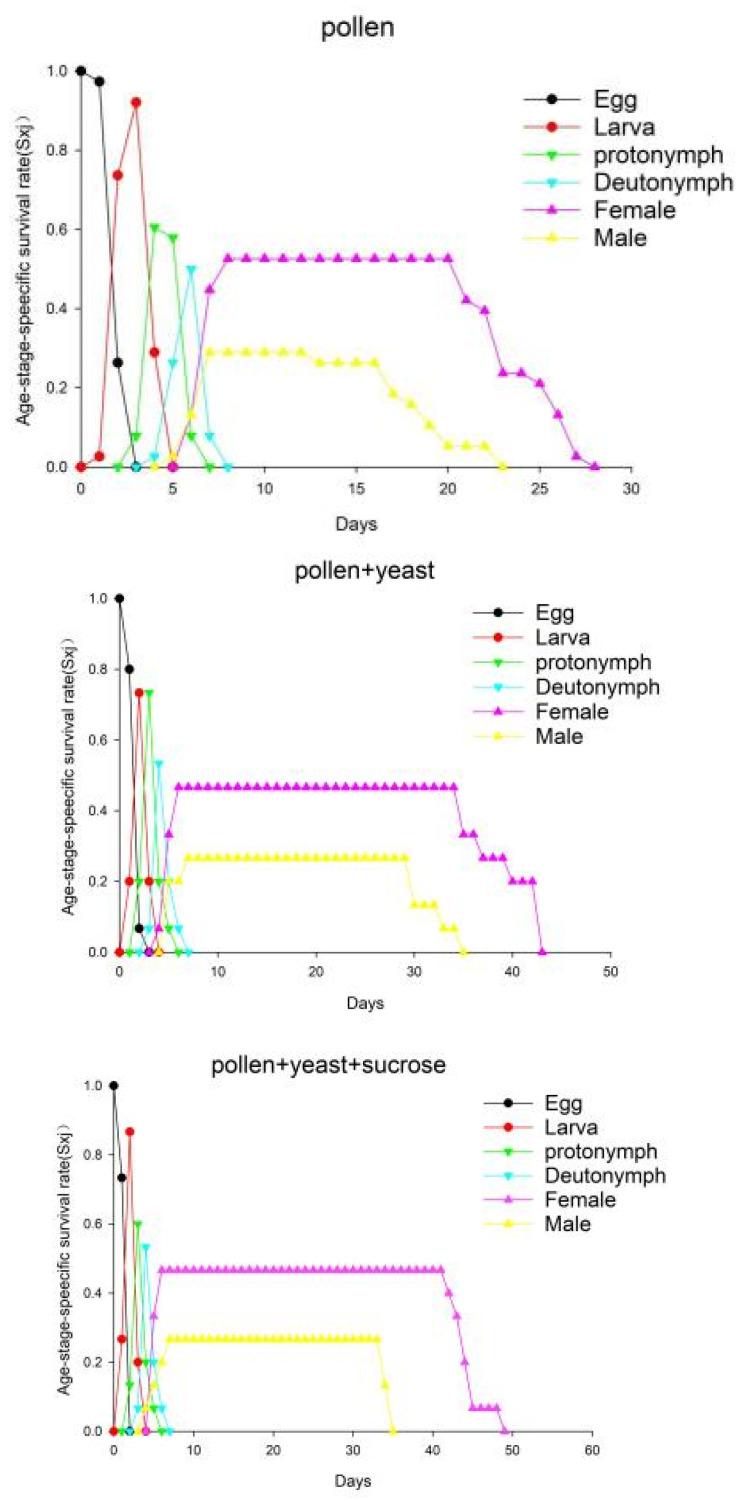
The survival rate of *Amblyseius orientalis* individuals in the three treatment groups. S_xj_ indicates the probability that a newly laid egg will survive to age x and stage j.

**Figure 2 insects-13-00983-f002:**
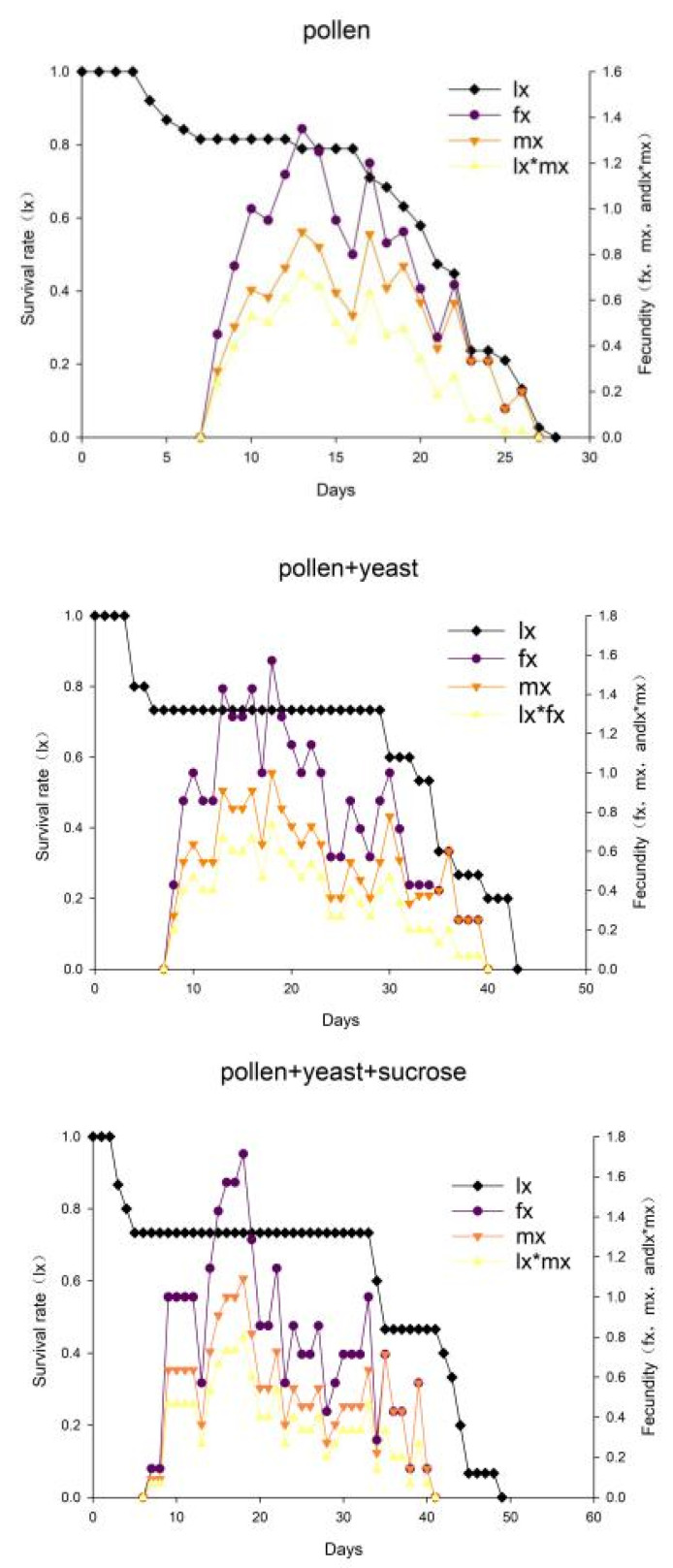
The population survival rate and fecundity of *Amblyseius orientalis* in the three treatment groups. l_x_ indicates the probability that a new egg will survive to age x; f_xj_ indicates the number of eggs laid by an adult female at age x and stage j; m_x_ indicates the mean fecundity of individuals at age x; l_x_*m_x_ indicates the product of l_x_ and m_x_, age-stage specific reproduction.

**Figure 3 insects-13-00983-f003:**
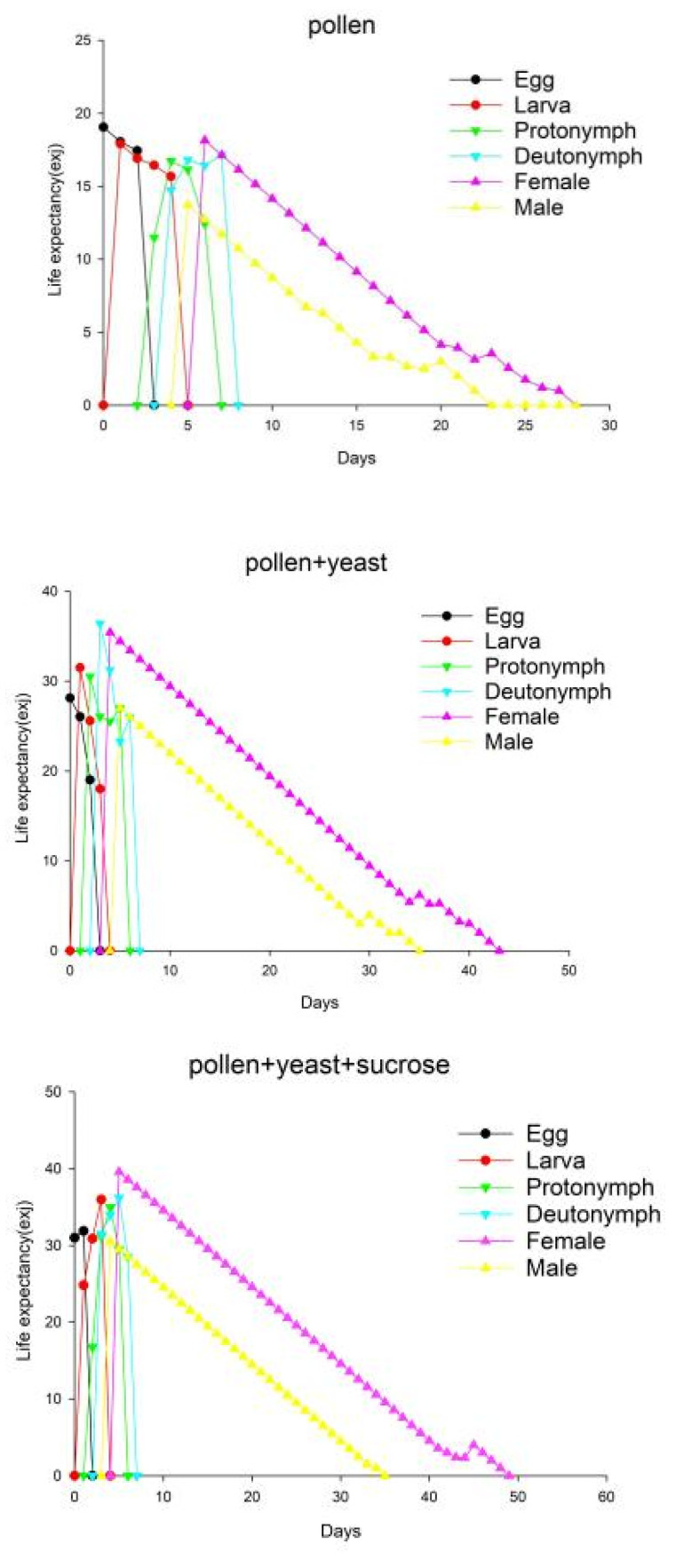
Life expectancy of *Amblyseius orientalis* individuals in three treatment groups; e_xj_ indicates the survival probability of an individual of age x and stage j.

**Figure 4 insects-13-00983-f004:**
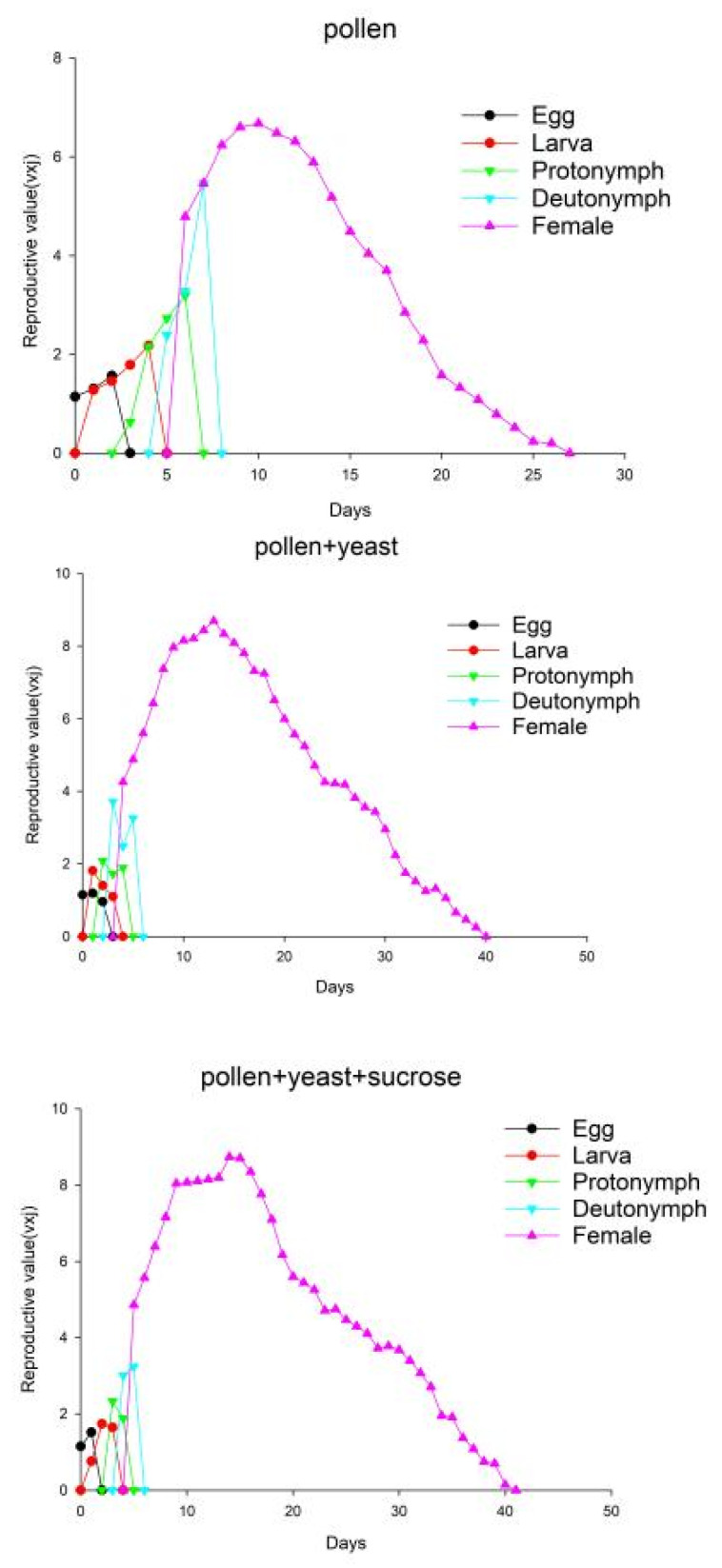
The reproductive value of *Amblyseius orientalis* individuals in the three treatment groups; v_xj_ indicates the contribution of an individual of age x and stage j to the future growth of the population.

**Figure 5 insects-13-00983-f005:**
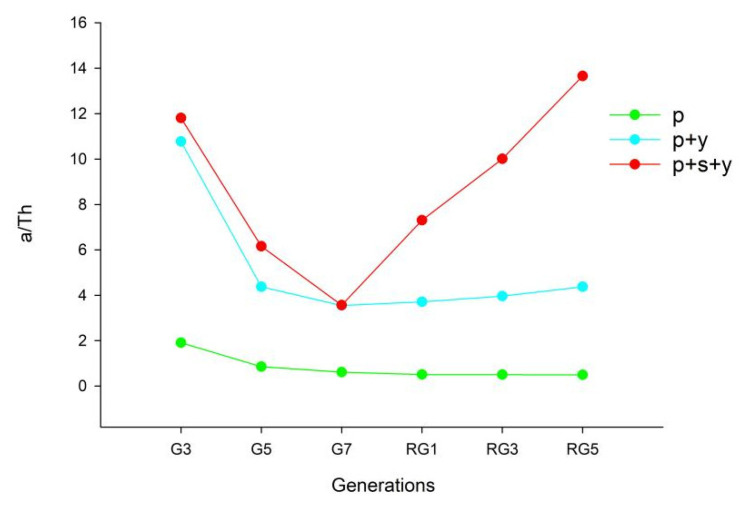
The effect of different treatments on predation capacity (a/Th).

**Figure 6 insects-13-00983-f006:**
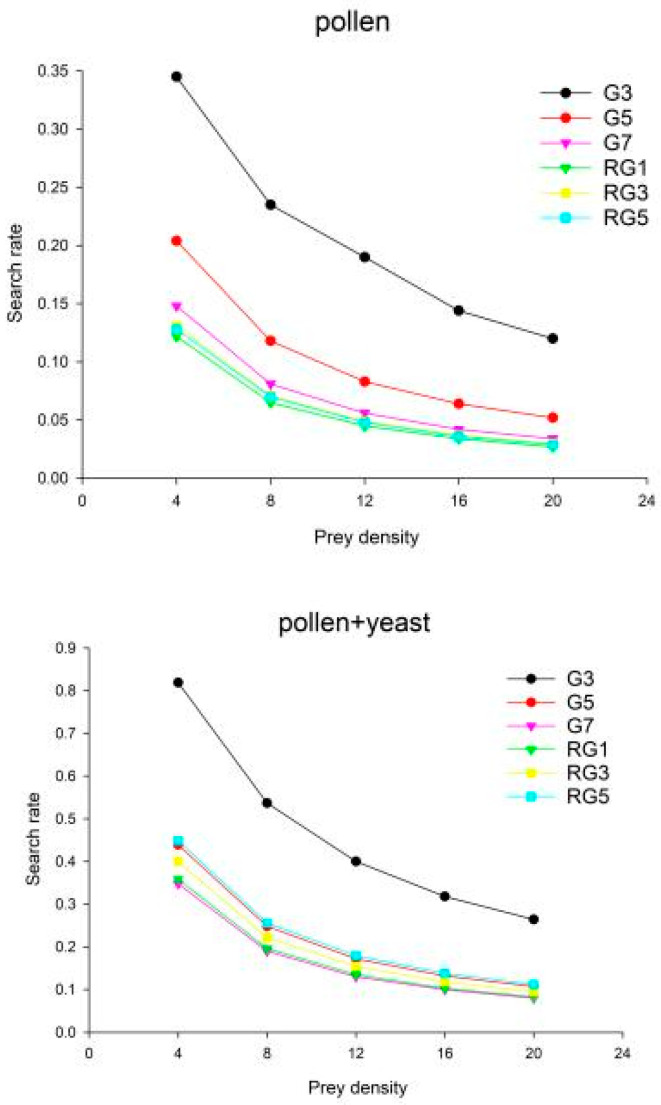
The search rate in the different treatment groups at different prey densities.

**Table 1 insects-13-00983-t001:** The length of each developmental stage of individuals with *Amblyseius orientalis* in the different treatment groups.

Duration, Days	Treatments		
	Pollen	Pollen + Yeast	Pollen + Yeast + Sucrose
Male			
Egg	2.09 ± 0.16 a	2.00 ± 0.01 a	1.75 ± 0.15 b
Larva	1.91 ± 0.16 a	1.25 ± 0.25 b	1.50 ± 0.29 b
Protonymph	1.45 ± 0.15 a	1.25 ± 0.15 b	1.25 ± 0.25 b
Deutonymph	1.00 ± 0.00 a	1.00 ± 0.01 a	1.00 ± 0.02 a
Preadult	6.45 ± 0.21 a	5.50 ± 0.50 b	5.50 ± 0.65 b
Adult longevity	12.27 ± 0.80 c	26.5 ± 1.55 b	29.00 ± 0.71 a
Total life span	18.73 ± 0.86 b	32.00 ± 1.22 a	34.50 ± 0.29 a
Female			
Egg	2.30 ± 0.11 a	1.57 ± 0.20 b	1.71 ± 0.18 b
Larva	2.05 ± 0.05 a	1.14 ± 0.10 c	1.43 ± 0.20 b
Protonymph	1.45 ± 0.11 a	1.43 ± 0.20 a	1.00 ± 0.01 b
Deutonymph	1.10 ± 0.07 a	1.00 ± 0.00 a	1.14 ± 0.14 a
Preadult	6.90 ± 0.14 a	5.14 ± 0.26 b	5.29 ± 0.18 b
Adult longevity	17.25 ± 0.51 c	34.29 ± 1.51 b	39.29 ± 0.87 a
Total life span	24.15 ± 0.54 c	39.43 ± 1.41 b	44.57 ± 0.84 a
All			
Egg	2.24 ± 0.08 a	1.87 ± 0.13 b	1.73 ± 0.12 b
Larva	2.00 ± 0.07 a	1.14 ± 0.09 b	1.36 ± 0.13 b
Protonymph	1.45 ± 0.09 a	1.33 ± 0.14 ab	1.08 ± 0.08 b
Deutonymph	1.06 ± 0.04 a	1.00 ± 0.01 a	1.09 ± 0.09 a
Preadult	6.74 ± 0.12 a	5.27 ± 0.24 b	5.36 ± 0.24 b
Adult longevity	15.48 ± 0.61 b	31.45 ± 1.59 a	35.55 ± 1.67 a
Total life span	19.05 ± 1.22 b	28.13 ± 3.95 a	31.00 ± 4.54 a

Note: Data in the table are represented as mean ± SE. The means followed by different letters in the same columns are significantly different at the 0.05 level based on one-way ANOVA and Tukey′s HSD multiple range test. “All” means”male + female”.

**Table 2 insects-13-00983-t002:** The population parameters of *Amblyseius orientalis* individuals in the different treatment groups.

Population Parameters	Treatments		
	Pollen	Pollen + Yeast	Pollen + Yeast + Sucrose
Net reproductive rate (R_0_)	7.11 ± 0.81 b	12.06 ± 1.31 a	12.60 ± 2.24 a
Intrinsic rate of increase (r/day)	0.13 ± 0.01 a	0.14 ± 0.02 a	0.14 ± 0.02 a
Finite rate of increase (λ/day)	1.14 ± 0.01 a	1.15 ± 0.02 a	1.15 ± 0.02 a
Gross reproduction rate (GRR)	10.53 ± 1.11 b	17.98 ± 2.56 a	18.11 ± 1.63 a
Mean generation time (T/day)	14.85 ± 0.19 b	18.13 ± 0.47 a	18.55 ± 0.52 a

Note: Data in the table are represented as mean ± SE. The means followed by different letters in the same columns are significantly different at the 0.05 level based on one-way ANOVA and Tukey′s HSD multiple range test.

**Table 3 insects-13-00983-t003:** The biological parameters of *Amblyseius orientalis* individuals in the different treatment groups.

Biological Parameters	Treatments		
	Pollen	Pollen + Yeast	Pollen + Yeast + Sucrose
Adult pre-oviposition period (APOP/day)	1.80 ± 0.11 a	3.42 ± 0.29 b	3.43 ± 0.29 b
Total pre-oviposition period (TPOP/day)	8.70 ± 0.16 a	8.57 ± 0.03 a	8.71 ± 0.11 a
Oviposition period (day)	11.20 ± 0.44 b	21.86 ± 0.61 a	23.14 ± 0.65 a
Fecundity (eggs/female)	13.50 ± 0.55 b	25.86 ± 0.80 a	27.00 ± 0.62 a
Female ratio of offspring	0.63 a	0.60 a	0.61 a

Note: Data in the table are represented as mean ± SE. The means followed by different letters in the same columns are significantly different at the 0.05 level based on one-way ANOVA and Tukey′s HSD multiple range test.

**Table 4 insects-13-00983-t004:** The functional response of *Amblyseius orientalis* to *Panonchus citri* in the different treatment groups.

Treatment	Generation	Instant Attack Rate	Handling Time	Predation Capacity	Daily Maximum Predation Number	Holling Disc Equation	
		a	Th(d)	a/Th	1/Th	Na = aN/(1 + aThN)	R²
Pollen	G3	0.649	0.340	1.909	2.941	Na = 0.649N/(1 + 0.220N)	0.958
	G5	0.769	0.900	0.854	1.111	Na = 0.769N/(1 + 0.692N)	0.974
	G7	0.862	1.400	0.616	0.714	Na = 0.862N/(1 + 1.207N)	0.942
	RG1	0.909	1.780	0.511	0.562	Na = 0.909N/(1 + 1.618N)	0.954
	RG3	0.807	1.600	0.504	0.625	Na = 0.807N/(1 + 1.291N)	0.988
	RG5	0.806	1.640	0.491	0.610	Na = 0.806N/(1 + 1.322N)	0.950
Pollen + Yeast	G3	1.724	0.160	10.775	6.250	Na = 1.724N/(1 + 0.276N)	0.989
	G5	1.923	0.440	4.370	2.273	Na = 1.923N/(1 + 0.846N)	0.986
	G7	2.128	0.600	3.547	1.667	Na = 2.128N/(1 + 1.277N)	0.979
	RG1	2.151	0.580	3.709	1.724	Na = 2.151N/(1 + 1.248N)	0.962
	RG3	1.980	0.500	3.960	2.000	Na = 1.980N/(1 + 0.990N)	0.982
	RG5	1.835	0.420	4.370	2.381	Na = 1.835N/(1 + 0.771N)	0.976
Pollen + Yeast + Sucrose	G3	1.653	0.140	11.807	7.143	Na = 1.653N/(1 + 0.232N)	0.979
	G5	1.724	0.280	6.157	3.571	Na = 1.724N/(1 + 0.482N)	0.987
	G7	1.852	0.520	3.562	1.923	Na = 1.852N/(1 + 0.963N)	0.975
	RG1	1.754	0.240	7.308	4.167	Na = 1.754N/(1 + 0.421N)	0.966
	RG3	1.802	0.180	10.011	5.555	Na = 1.802N/(1 + 0.324N)	0.974
	RG5	1.639	0.120	13.658	8.333	Na = 1.639N/(1 + 0.197N)	0.969

**Table 5 insects-13-00983-t005:** The search rate of *Amblyseius orientalis* individuals in the different treatment groups.

Treatment	Generation	Searching Effect						Linear Regression Equation	R
Pollen	G3	Density	4	8	12	16	20	y = 0.3691 − 0.0135x	0.925
			0.345	0.235	0.190	0.144	0.120		
	G5	Density	4	8	12	16	20	y = 0.2116 − 0.009x	0.858
			0.204	0.118	0.083	0.064	0.052		
	G7	Density	4	8	12	16	20	y = 0.1523 − 0.0067x	0.843
			0.148	0.081	0.056	0.042	0.034		
	RG1	Density	4	8	12	16	20	y = 0.1249 − 0.0055x	0.835
			0.122	0.065	0.045	0.034	0.027		
	RG3	Density	4	8	12	16	20	y = 0.1344 − 0.0059x	0.838
			0.131	0.071	0.049	0.037	0.030		
	RG5	Density	4	8	12	16	20	y = 0.1318 − 0.0058x	0.844
			0.128	0.070	0.048	0.036	0.029		
Pollen + Yeast	G3	Density	4	8	12	16	20	y = 0.8663 − 0.0332x	0.898
			0.819	0.537	0.400	0.318	0.264		
	G5	Density	4	8	12	16	20	y = 0.4536 − 0.0195x	0.850
			0.439	0.248	0.172	0.132	0.107		
	G7	Density	4	8	12	16	20	y = 0.3574 − 0.0157x	0.840
			0.348	0.190	0.130	0.100	0.080		
	RG1	Density	4	8	12	16	20	y = 0.3679 − 0.0161x	0.841
			0.358	0.196	0.135	0.103	0.083		
	RG3	Density	4	8	12	16	20	y = 0.412 − 0.0179x	0.845
			0.400	0.222	0.154	0.118	0.095		
	RG5	Density	4	8	12	16	20	y = 0.4644 − 0.0198x	0.853
			0.449	0.256	0.179	0.138	0.112		
Pollen + Yeast + Sucrose	G3	Density	4	8	12	16	20	y = 0.9102 − 0.0339x	0.907
			0.857	0.579	0.437	0.351	0.293		
	G5	Density	4	8	12	16	20	y = 0.6149 − 0.0253x	0.870
			0.589	0.355	0.254	0.198	0.162		
	G7	Density	4	8	12	16	20	y = 0.3932 − 0.017x	0.848
			0.381	0.213	0.148	0.113	0.091		
	RG1	Density	4	8	12	16	20	y = 0.6851 − 0.0278x	0.877
			0.654	0.402	0.290	0.227	0.186		
	RG3	Density	4	8	12	16	20	y = 0.8273 − 0.0325x	0.890
			0.785	0.502	0.369	0.291	0.241		
	RG5	Density	4	8	12	16	20	y = 0.9767 − 0.0353x	0.915
			0.917	0.636	0.487	0.395	0.332		

## Data Availability

Data may be requested from the corresponding authors.

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
