# Peer review of "Age-Stage, Two-Sex Life Table and Functional Response of Amblyseius orientalis (Acari: Phytoseiidae) Feeding on Different Nutrient Sources"

_insects, 2022, doi:10.3390/insects13110983_

Round 1
Reviewer 1 Report
This manuscript presents the results of a study on the Age-stage, two-sex life table and functional response of Amblyseius orientalis (Acari: Phytoseiidae) feeding on different nutrient source. The results could be interesting for readers and could supplement existing literature and is acceptable after minor suggestion. My specific comments and corrections can be found in attached pdf.

Author Response
Dear reviewer, I have revised the manuscript according to your suggestions and requirements, please see the attachment

Reviewer 2 Report
Review of the MS: Age-stage, two-sex life table and functional response of Amblyseius orientalis (Acari: Phytoseiidae) feeding on different nutrient sources
The MS presents the results of laboratory experiments on a predatory mite, Amblyseius orientalis to determine its growth, development, reproduction, and functional response, when different food types were used (pollen, pollen + yeast, and pollen + yeast + sucrose).
The study is valuable, but ms should be better written. Introduction should be corrected and study aim should be added. Methods should be described more clearly. Results include references (in Table 1 and in the text), suggesting that these are not results of the experiment but from the literature – it must be clarified. Results include parameters and indices not mentioned in the Methods – they all should be clarified in the Methods. Not all results are also reflected in the Abstract. Discussion is repeating too much the Results, but is not discussing them well with the literature. References follow strange formatting and should be adjusted to the Journal requirements.
Additional comments are included in the text. The text is generally understandable but the ms would benefit from editing by the language expert or native speaker.
In summary, after major revisions the ms can be considered for publication in the Insects.

Author Response

(The authors gave the same response as above.)

Reviewer 3 Report
This is an excellent paper providing solutions for the very important rearing of predatory mites. I have already sent the manuscript with a few corrections/ suggestions.

Author Response

(The authors gave the same response as above.)

Round 2
Reviewer 2 Report
Dear Authors,
you improved the MS, but still minor corrections are needed, and especially extensive editing of language and style is required.
Kind regards

Author Response
Dear reviewer, I have revised the manuscript according to your suggestions and requirements.The manuscript has been checked by a native speaker,please see the attachment.
